# Peroxisome Proliferator-Activated Receptor Alpha Stimulation Preserves Renal Tight Junction Components in a Rat Model of Early-Stage Diabetic Nephropathy

**DOI:** 10.3390/ijms252313152

**Published:** 2024-12-07

**Authors:** Lorena Rosas-Martínez, Rafael Rodríguez-Muñoz, María del Carmen Namorado-Tonix, Fanis Missirlis, Leonardo del Valle-Mondragón, Alicia Sánchez-Mendoza, José L. Reyes-Sánchez, Luz Graciela Cervantes-Pérez

**Affiliations:** 1Department of Pharmacology, National Institute of Cardiology Ignacio Chávez, Juan Badiano No. 1, Col. Seccion XVI, Tlalpan, Mexico City 14080, Mexico; lor_ros@fisio.cinvestav.mx (L.R.-M.); leonardodvm65@hotmail.com (L.d.V.-M.); masanchez@gmail.com (A.S.-M.); 2Department of Physiology, Biophysics, and Neuroscience, Center for Research and Advanced Studies of National Polytechnic Institute, CINVESTAV-IPN, Instituto Politécnico Nacional 2508, San Pedro Zacatenco, Gustavo A. Madero, Mexico City 07360, Mexico; rafaelr@fisio.cinvestav.mx (R.R.-M.); cnamora17@hotmail.com (M.d.C.N.-T.); fanis@fisio.cinvestav.mx (F.M.); jreyes@fisio.cinvestav.mx (J.L.R.-S.)

**Keywords:** claudin, hyperglycemia, fibrates, PPAR-α, diabetic nephropathy

## Abstract

Chronic hyperglycemia results in morphological and functional alterations of the kidney and microvascular damage, leading to diabetic nephropathy (DN). Since DN progresses to irreversible renal damage, it is important to elucidate a pharmacological strategy aimed for treating DN in the early stage. Here, we used the type 2 diabetic rat model to induce DN and show a nephroprotective effect following the stimulation of PPAR-α, which stabilized renal tight junction components claudin-2, claudin-5, and claudin-16. At 14 weeks old, streptozotocin-induced DN, evidenced by elevated creatinine clearance, proteinuria, and electrolyte excretion, was followed by an elevation in oxidative stress and increasing MMP activities affecting the integrity of claudin-2 and claudin-5. Treatment with a PPAR-α agonists decreased glucose levels in diabetic rats. In addition, we found that the expressions of CLDN-5 in glomeruli, CLDN-2 in proximal tubules, and CLDN-16 in the thick ascending limb of the loop of Henle were increased after treatment. As a result, renal function improved, while the oxidative stress and enzymatic activity of MMP-2 and MMP-9 decreased. In conclusion, PPAR-α stimulation prevented the decrease in claudins through a mechanism involving a correction of hyperglycemia, decreasing it in kidney oxidative stress and MMP-2 and MMP-9 activities, showing a promising nephroprotective action in the early stage of DN.

## 1. Introduction

Diabetes mellitus (DM) is an important global health issue, and its prevalence is increasing rapidly around the world. This syndrome is characterized by high glucose levels (hyperglycemia) due to a deficiency in the production or function of insulin or both [1]. Uncontrolled and chronic hyperglycemia can damage major organs, including the kidney. One of the main complications is diabetic nephropathy (DN), and it occurs in 20–50% of diabetic patients [2]. DN is associated with morphological alterations that can result in functional alterations such as proteinuria, an increase in creatinine clearance, and higher fractional excretions of sodium, potassium, magnesium, and calcium [3,4,5]. We have recently demonstrated that these alterations are related to oxidative stress and increased activity of matrix metalloproteinases (MMPs) that affect the integrity of claudin-5 (CLDN-5) in glomeruli and claudin-2 (CLDN-2) in proximal tubules by promoting their degradation due to post-translational modifications [6]. 

Claudins are transmembrane proteins that form tight junctions (TJs) and are expressed in epithelium and endothelium, being highly relevant in the kidney. They form paracellular barriers and pores that determine the permeability of the kidney to small ions, solutes, and water [7,8]. There is no current treatment that prevents the loss of claudins in early stages of DN.

Peroxisome proliferator-activated receptor alpha (PPAR-α) is a member of the nuclear hormone receptor superfamily of ligand-activated transcription factors. PPAR-α is abundantly expressed in the proximal tubules, the medullary thick ascending limbs, and, to a lesser extent, in the glomerular mesangial cells. PPAR-α agonists, such as fibrates, are synthetic ligands that promote PPAR-α activation and thus induce the transcription of genes involved in the β-oxidation of fatty acids [9,10,11]. Currently, experimental and clinical evidence suggests that PPAR-α is involved in the pathogenesis of DN, and it contributes to the metabolic control of renal function, showing a nephroprotective effect [12,13,14,15]. Nevertheless, the mechanism of the beneficial effects of PPAR-α agonists on the kidney in the early stages of DN has not been fully explored and may include non-metabolic mechanisms such as glucose or lipid handling. The present study has been designed to investigate the effect of PPAR-α activation (clofibrate) in treating DN and its effects on renal claudins in a type 2 diabetic (DM2)-rat model.

## 2. Results

### 2.1. PPAR-α Agonist Decreases Hyperglycemia but Does Not Prevent Body Weight Loss in Type 2 Diabetic Rats

The hallmark changes after the induction of diabetes with STZ in rats are the loss in body weight and the increase in blood glucose levels. Therefore, our first aim was to determine the effect of PPAR-α agonists on these parameters at 14 weeks. As shown in Figure 1A, the body weight of the DM2 group without treatment was reduced on average by 18%, compared to that of the CTRL. The PPAR-α agonists did not prevent the loss of body weight compared with the DM2 group. However, we found that the PPAR-α agonists decreased hyperglycemia (from 273.4 mg/dL to 159.4 mg/dL), A1C (from 12.9% to 5.6%), and glucose tolerance in DM2-treated rats (Figure 1B–E). These findings indicate that the PPAR-α agonists improved blood glucose levels without affecting body weight and serum insulin levels.

### 2.2. PPAR-α Agonist Treatment Improves Diabetes-Induced Renal Dysfunction and Nephrin Expression

The next step was to evaluate the effect of PPAR-α agonists on renal glomerular and tubular dysfunction induced by diabetes. Early markers of renal dysfunction at the glomerular level in DN have been reported as the presence of proteinuria (~2-fold), increased creatinine clearance (~2-fold), and decreased podocyte-specific proteins such as nephrin [16]. Therefore, we evaluated these markers in the absence or presence of PPAR-α agonists. As shown in Table 1, treatment with PPAR-α agonists significantly decreased proteinuria (from ~2-fold to 1.5-fold) and creatinine clearance (from ~2-fold to 1.3-fold) compared to those in the DM2 group.

Nephrin expression in renal glomeruli was measured by immunofluorescence (Figure 2A–D) and Western blot assays (Figure 2E). As expected, a significant decrease in nephrin expression was found in the DM2 group compared with that in the CTRL group, which was significantly improved by PPAR-α agonist treatment.

In addition, indicators of tubular dysfunction include increased fractional excretion of ions such as sodium, potassium, calcium, and magnesium. In our model, at 14 weeks of life, we observed an increase in the fractional excretion of these ions in diabetic rats. However, we observed that when diabetic rats were treated with PPAR-α agonists, the fractional excretion values were not increased (Table 1). These results indicate that the PPAR-α agonists exhibited a nephroprotective response in both renal segments.

### 2.3. Diabetes Induces a Decrease in PPAR-α Activity and Treatment with a PPAR-α Agonist Induces Activation of This Receptor in the Kidney of Type 2 Diabetic Rats

Once the pathophysiological data of the diabetes model and renal function were revealed after PPAR-α activation; the next step was to evaluate the expression, localization, and activity of PPAR-α in two nephron segments after clofibrate (100 mg/kg) administration. Our data show that PPAR-α expression decreases in the glomeruli (Figure 3C) of diabetic rats in contrast with that in their control (Figure 3A); similarly, the activity of this receptor decreases ~0.5 fold in diabetic conditions (Figure 3K). In comparison, the stimulation of PPAR-α agonists increases its expression (Figure 3D -glomeruli-) and activity (1.5-fold, Figure 3K) in the kidneys of type 2 diabetic rats. Clofibrate treatment did not change the PPAR-α expression in the control rats (Figure 3B -glomeruli- and Figure 3F -proximal tubules-) but increased its activity (Figure 3K). The Western blot assay and densitometric analysis for isolated glomeruli and isolated proximal tubules are shown in Figure 3I,J.

### 2.4. PPAR-α Agonist Treatment Improves Claudin-2, Claudin-5, and Claudin-16 Expressions in Proximal Tubules, Glomeruli, and Thick Ascending Limb of the Loop of Henle, Respectively

From our previously reported work, we knew that an increase in oxidative stress and MMP activities affects the integrity of renal tight junction (TJ) proteins such as claudin-2 and claudin-5 in the early stage of DN [6]. Since our results showed a decrease in these mechanisms after treatment with PPAR-α agonists, we evaluated the expression of TJ proteins in the kidneys of diabetic rats in the presence and absence of PPAR-α agonists. Immunofluorescence and Western blot assays of claudin-5 (Figure 4A–E) and claudin-2 (Figure 4F,J) were performed. PPAR-α activation improved the claudin-2 and claudin-5 expressions evaluated by immunofluorescence and Western blot analysis from isolated proximal tubules and glomeruli, respectively.

In addition, due to increases in the fractional excretions of calcium and magnesium (Table 1), we evaluated the expression of claudin-16 by immunofluorescence. We found that the expression of this claudin, which is important for the regulation of the reabsorption of these divalent ions in the kidney, decreased in the TAL of untreated diabetic rats (Figure 5C), whereas this was not observed in treated diabetic rats (Figure 5D).

### 2.5. Diabetes-Induced Oxidative Stress and Activity of MMP-2 and MMP-9 in the Kidney Were Improved by PPAR-α Agonist Treatment

Among the mechanisms related to the nephroprotective effect of PPAR-α agonists are the decreases in oxidative stress and the activity of MMPs in the kidney [9,17,18]. Therefore, we evaluated the antioxidant effect of PPAR-α activation in the glomeruli and proximal tubules of diabetic rats by measuring oxidative stress markers. As illustrated in Figure 6, hyperglycemia resulted in a decreased antioxidant capacity (25% decrease in glomeruli and 28% decrease in proximal tubules) (Figure 6A,C) and increased lipoperoxidation (1.5-fold in glomeruli and ~3-fold in proximal tubules) (Figure 6B,D) in the kidneys of DM2 rats. Instead, PPAR-α agonist treatment had an antioxidant effect, evidenced by increasing antioxidant capacity (50% increase in glomeruli and 30% increase in proximal tubules) and decreasing MDA levels (~40% decrease in glomeruli and ~85% decrease in proximal tubules) in DM2 rats.

The next step was to evaluate the renal activity of MMPs by gelatin zymography assay. Gel zymography and densitometric analysis revealed that the active forms of MMP-2 (62 kDa) and MMP-9 (82 kDa) in glomeruli (Figure 7A–C) as well as proximal tubules (Figure 7D–F) of DM2 were elevated (~1.5-fold) compared with those in the CTRL rats. Treatment with PPAR-α agonists decreased this activity in diabetic rats in both nephron segments.

Our data suggest that PPAR-α activation enhances renal claudin protein levels in diabetic rats by decreasing oxidative stress and MMP activity, mechanisms associated with claudin degradation.

The results at the level of protein expression of MMP-2 and -9 in the kidney by immunofluorescence showed the same effect as that of PPAR-α agonists on their activity in glomeruli and proximal tubules (MMP-2 Appendix A and MMP-9 Appendix A). These results demonstrate that the nephroprotective role of PPAR-α agonist treatment in our diabetic model is through attenuation of the oxidative stress and the activity of MMPs that promote the development of DN.

## 3. Discussion

We previously reported that hyperglycemia produced damage to claudins, renal TJ proteins expressed in glomeruli, and proximal tubules, the two major targets of renal damage in early DN [6,19]. So far, we know that the loss of claudins in those kidney segments is due to two mechanisms of damage: increased oxidative stress and increased MMP activities. The aim of this study was to explore whether those alterations in claudins might be ameliorated by PPAR-α agonists, based on the knowledge of their nephroprotective role through an antioxidant effect and a decrease in the activity of MMPs [9,17,18].

Several studies provide evidence that PPAR-α plays an important role in renal homeostasis and is involved in many renal pathophysiological conditions, including DN; therefore it has been proposed as a promising therapeutic target for treating this disease [20]. Few studies have been reported on the expression of this receptor in the diabetic condition; one of them evaluated the expression of the three identified isoforms of PPARs in type 1 and 2 diabetic models. Interestingly, it was found that PPAR-α, but not PPAR-β or PPAR-γ, is downregulated in both models and that hyperglycemia is a direct cause of PPAR-α downregulation [21]. Another experimental study reported that diabetes-induced kidney damage is more severe in PPARα-knockout diabetic mice [22]. In clinical studies, Herman et al. reported that PPAR-α was downregulated along with other genes involved in fatty acid oxidation (FAO) pathways in kidney biopsies from DN patients [23]. In our experimental study, we observed a decrease in the kidney expression and activity of PPAR-α in a rat model of type 2 diabetes. It is worth mentioning that in comparison with previous reports by other groups, we are the first to show the decreases in PPAR-α expression and activity in extracts of glomeruli and proximal tubules, making our evaluation more specific. In addition, our data showed that daily treatment with a PPAR-α agonist for 2 weeks was sufficient to prevent the loss of kidney expression and increase the activity of renal PPAR-α in the diabetic condition.

According to the literature, PPAR-α once activated regulates a set of genes essential for the β-oxidation of fatty acids, improving renal metabolic control. Furthermore, PPAR-α has been reported to be involved in the interaction between hyperglycemia and dyslipidemia in DN [24,25]. Park et al. reported that the improvement of glucose handling by fenofibrate, a PPAR-α agonist, is a mechanism to prevent the progression of DN [26]. Therefore, we evaluated glucose handling with and without PPAR-α treatment in diabetic rats. We found that the PPAR-α agonist improved glucose metabolism as evidenced by reduced blood glucose, A1C, and glucose tolerance. Thus, exogenous treatment with a PPAR-α agonist might improve renal homeostasis in the early stages of DN in our model of type 2 diabetes. Additionally, Gonzalez, F.J. et al. reported that animal models expressing human PPAR-α stimulated with proper agonists expressed Let-7c miRNA [27]. Interestingly this miRNA is involved in the regulation of insulin response, glucose homeostasis, lipid metabolism, and oxidative injury, which may represent another pathway of renal protection.

The proposal that PPAR-α has a nephroprotective function was reinforced by the results of proteinuria, creatinine clearance, and the fractional excretion of ions secondary to treatment. In agreement with these findings, glomerular damage as assessed by nephrin expression was attenuated by the PPAR-α agonist. Some studies have suggested that PPAR-α has nephroprotective properties due to the decrease in oxidative stress and activity of MMPs in the kidney [9,17,18]. Our results showed that glomerular and proximal tubule extracts from rats that did not receive PPAR-α agonist treatment exhibited increased oxidative stress and MMP-2 and MMP-9 activity. By contrast, the PPAR-α agonist attenuated oxidative stress and lipoperoxidation in both renal extracts. Indeed, the expression and activity of MMPs was also decreased in the glomeruli and proximal tubules of treated diabetic rats. These data suggest that the nephroprotective effect of PPAR-α is through the regulation of these two mechanisms in DN.

The relevance of our study was to evaluate the renal expression of claudins in the presence and absence of a PPAR-α agonist because it has recently been suggested that TJ proteins play an important role in the pathogenesis of DN [6,19,28]. Furthermore, we reported that claudin-2 expression in proximal tubules and claudin-5 expression in glomeruli were significantly reduced in DN, leading to glomeruli permeability and decreased ion reabsorption in the proximal tubule in the early state of DN. In addition, we found that renal oxidative stress and MMP activities in DN promote the degradation of these claudins [6]. Therefore, we hypothesized that treatment with a PPAR-α agonist by decreasing these mechanisms might attenuate the renal expression of these TJ proteins and thus improve renal function in type 2 diabetic rats. According to our results, the PPAR-α agonist improved claudin-5 expression in glomeruli, suggesting a possible association with decreased proteinuria. In a similar way, the PPAR-α agonist improved claudin-2 expression in proximal tubules. This effect might be closely related to renal sodium and potassium handling through the paracellular pathway [7,29,30].

In order to explain the increase in FEMg, an immunofluorescence assay of claudin-16 from kidney tissue was performed. Claudin-16 is expressed in the thick ascending limb of the loop of Henle (TAL), and it is important for the reabsorption of magnesium and calcium [31]. Therefore, it has been reported that downregulation of claudin-16 might be the cause of hypermagnesuria and hypercalciuria [32]. Interestingly, our results showed that claudin-16 expression was decreased in diabetic rats and that the PPAR-α agonist raised the expression of this claudin in TAL. Regarding renal calcium handling, it has been proposed that claudin-2 expression in the proximal tubule could be participating in its paracellular transport [33,34]. For that reason, the improvement in the handling of this cation could be due to the prevention of the loss of claudin-2 after PPAR-α agonist treatment. To our knowledge, this is the first study that evaluates the regulation of claudin-16 expression by PPAR-α agonists. Thus, future work is required to evaluate how PPAR-α agonists participate in claudin-16 expression regulation.

We acknowledge some limitations in our study. For instance, it is well-known that rodents may exhibit a hepatotoxic response under prolonged administration of certain PPAR-α agonists, once the treatment is prolonged [35]. Being basic research as the first step addressing pharmacological responses, intended mechanisms of action, and adverse responses, it would be relevant to explore these findings in an experimental model of rodents expressing humanized PPAR-α.

## 4. Materials and Methods

### 4.1. Reagents

The mouse anti-nephrin, rabbit anti-PPAR-α, rabbit anti-MMP-2, and rabbit anti-MMP-9 were purchased from Santa Cruz Biotech, Inc. (Santa Cruz Biotechnology Inc., Santa Cruz, CA, USA). The rabbit anti-claudin-2 and rabbit anti-claudin-5 primary antibodies, the peroxidase-conjugated anti-rabbit, peroxidase-conjugated anti-mouse, Alexa Fluor^®^ 488 donkey anti-rabbit, Alexa Fluor^®^ 488 donkey anti-goat, and Alexa Fluor^®^ 594 donkey anti-mouse secondary antibodies were purchased from Invitrogen (ThermoFisher Scientific, Carlsbad, CA, USA). The rabbit anti-claudin-16 primary antibody was purchased from Abcam (Abcam, Waltham, MA, USA). The mouse anti-dipeptidylpeptidase-IV (DPP4) primary antibody was purchased from AbD serotec (Bio-Rad, Hercules, CA, USA). Percoll, streptozotocin (STZ), ammonium ferrous sulfate, methanol, bovine serum albumin (BSA), anti β-actin-peroxidase antibody, and all other reagents were purchased from Sigma-Aldrich Co. (Sigma-Aldrich Co., Saint Louis, MO, USA) unless otherwise indicated.

### 4.2. Experimental Design

All animal procedures were performed in accordance with Mexican Federal Regulations concerning Animal Experimentation and Care (Ministry of Agriculture, SAGARPA, NOM-062-ZOO-1999, Mexico). The animal protocol was approved by the Ethical Committee of the National Institute of Cardiology Ignacio Chávez in Mexico City and conducted according to the Guidelines for Care and Use of Experimental Animals (Protocol INC/CICUAL/004/2020) and the U.S. National Institutes of Health guide for the care and use of Laboratory animals.

To estimate the sample size, we used the equation proposed by Rojo-Amigo, A. [36]. This formula calculates the minimum number of animals required to carry out an experiment and warrant statistical power. Since all animals were under the same conditions, there was no need to adjust for any variables. To fulfill the 3R’s rule, we began the protocol assuming a 0% of death secondary to streptozotocin or vehicle administration. The presumption was true; therefore, the number of animals included at the beginning of the study was the same at the end. The formula is as follows:X = N/((A/199) × (B/100) × (C/100) × …)
where:
X = represents total number of experimental subjects required for the study;N = statistical minimum number, which can be found in the published literature or with statistical calculations. In previous reports from our laboratory, the sample size used for this model has been 6–9 animals per group [6,37];A = 100-% of incidence 1 (e.g., death following streptozotocin administration);B = 100-% of incidence 2, and so on.

To establish the type 2 diabetes mellitus model, neonate (2 days old) male Wistar rats were first divided into 2 groups: (1) Control group (CTRL) received 0.1 M citrate buffer, pH 4.5 (vehicle), intraperitoneally (i.p.) and (2) DM2 group received a single STZ dose (70 mg/kg, i.p.) dissolved in vehicle. Twelve weeks after STZ administration, animals from both experimental groups were randomly subdivided (Appendix A): (a) CTRL + vehicle-treated, (b) CTRL + PPAR-α agonist (clofibrate, 100 mg/kg), (c) DM2 + vehicle-treated, and (d) DM2 + PPAR-α agonist (clofibrate, 100 mg/kg). Previously, we carried out a dose–response curve with clofibrate as a pharmacological tool to specifically stimulate PPAR-α, and we reported that 100 mg/kg i.p./day is enough to stimulate PPAR-α [38]. The rats from the four groups were allocated in metabolic cages at 14 weeks of life to collect 24 h urine for measurement of proteinuria, creatinine, and electrolytes. Animals were euthanized by exposing them to an isoflurane overdose (Sofloran, PiSA Pharmaceutical, Guadalajara, Jal., Mexico), and their kidneys were excised and weighed.

### 4.3. Physiological and Renal Function Parameters

Body weight and blood glucose levels were measured on a weekly basis after STZ administration. Blood from the tail was collected for capillary glucose determination in non-fasting rats using a glucometer (Accu-Chek Active, Glucotrend, Roche Diagnostics, Mannheim, Germany). Glycated hemoglobin (A1C) values were obtained using a kit (A1CNow-plus, PTS Diagnostics, Indianapolis, IN, USA). Rats from every group, 14 weeks after vehicle- or STZ administration, were fasted for 14 h and subjected to an oral glucose tolerance test (OGTT). Briefly, a 2 g/kg glucose load was orally administrated, and tail capillary glucose was measured with Accu-Chek monitor (Roche Diagnostics, Mannheim, Germany) at 0 (before glucose load), 30, 60, 90, and 120 min after glucose [6]. We determined insulin secretion by enzyme-linked immunosorbent assay (rat ELISA Kit, EMD Millipore, Darmstadt, Germany). On the following day, under anesthesia with isoflurane 3% (Sofloran, PiSA Pharmaceutical, Guadalajara, Jal., Mexico), blood was collected by cardiac puncture and serum was separated and stored at −70 °C for further analysis.

Total urinary protein was measured by the Lowry method (Bio-Rad Protein Assay Kit, Bio-Rad Laboratories, Hercules, CA, USA). Urinary and serum creatinine were measured by the modified Jaffé reaction, as previously described [39]. Creatinine clearance (CCr) was calculated by a standard formula. CCr = [Urine creatinine (mg/mL)] × urine flow (mL/min)/[Serum creatinine (mg/mL)]. The results are reported as mL/min/100 g body weight (BW).

Urinary glucose was estimated with Mission Urinalysis Reagent Strips (ACON Laboratories, San Diego, CA, USA). Urinary and serum electrolytes concentrations were measured, as previously reported, by inductively coupled plasma optical emission spectrometry (ICP-OES) in an Optima 8300 ICP-OES (Perkin Elmer Inc., Shelton, CT, USA) [6,40]. The concentrations of Na^+^, K^+^, Mg^2+^, and Ca^2+^ were measured against calibration curves and a digestion blank.

### 4.4. Isolation of Glomeruli

Glomeruli were isolated by the mechanical graded sieving technique as previously described [41]. The protein content in these glomeruli lysates was determined by the Lowry colorimetric method (Bio-Rad Protein Assay Kit, Bio-Rad Laboratories, Hercules, CA, USA).

### 4.5. Isolation of Proximal Tubules

Renal tubules were isolated from the renal cortex by Percoll density gradient centrifugation as previously described [42].

### 4.6. Immunofluorescence

Kidney samples were cryopreserved with 2-methylbutane. Kidney slices (6 μm of thickness) were obtained with a Leica Cm150 cryostat (Leica Biosystems, Wetzlar, Germany) and mounted on gelatin-coated slides and were kept frozen at −70 °C. The sections were fixed for 10 min with absolute methanol, and subsequently incubated for 5 min at room temperature (RT) in 1% (*v*/*v*) Triton X-100. Then, the tissue sections were washed 3 times with PBS, blocked for 1 h at RT with 1% (*w*/*v*) IgG-free albumin (1331-A, Research Organics Inc., Cleveland, OH, USA), and incubated overnight at 4 °C with one of the following primary polyclonal antibodies: anti-nephrin (1:100); anti-claudin-2, -5, -16 (1:100); PPAR-α (1:100); MMP-2 (1:50); MMP-9 (1:50); or monoclonal anti-DPP4 (1:300). Secondary antibodies Alexa Fluor 488 donkey anti-rabbit, Alexa Fluor 488 donkey anti-goat, and Alexa Fluor 594 donkey anti-mouse (1:300) were used. Immunofluorescence was evaluated using a confocal inverted microscope TCS-SP8 (Leica Microsystems, Heidelberg, Germany) and Leica Application Suite V. 3.7.4 (LAS AF LITE, Leica Microsystems, Heidelberg, Germany). Images were processed using the software ImageJ 3.3.0 (National Institutes of Health, Bethesda, MD, USA).

### 4.7. Western Blot Analysis

Protein samples (50 μg) were separated by SDS–PAGE gels and transferred to polyvinylidene difluoride (PVDF) membranes (Millipore Corp. Bedford, MA, USA). Nonspecific protein binding was blocked with casein 3× (sp-5020, VECTOR, Burlingame, CA, USA) for 1 h at RT. Membranes were incubated overnight at 4 °C with the appropriate primary antibodies (anti-nephrin (1:500), anti-claudin-2 and -5 (1:500), and anti-PPAR-α (1:250)). The proteins were detected using appropriate peroxidase-conjugated secondary antibodies (1:10,000) at RT, and chemiluminescence was detected (Chemidoc Workflow System, Bio-Rad Laboratories, Hercules, CA, USA). Quantification was performed by measurement of signal intensity with Image J software V. 3.3.0 (National Institute of Health, Bethesda, MD, USA).

### 4.8. Palmitoyl CoA Oxidase Assay

We measured the activity of peroxisome palmitoyl CoA oxidase, following the method previously described [20]. Renal cortex from all groups were homogenized (1:3 *w*/*v*) in 0.25 mol/L sucrose, 1 mmol/L EDTA, and 0.1% ethanol. Samples containing 500 μg of protein were incubated at 37 °C for 30 min, with the reaction mix containing 60 mmol/L Tris–HCl, 35 μmol/L palmitoyl CoA, 50 μmol/L FAD, 1 μmol/L scopoletin peroxidase (3 units), 0.6 mg bovine serum albumin, and triton X-100 (0.01%) to a final volume of 1 mL. The reaction was stopped by the addition of 4 mL of 0.1 mol/L borate buffer (pH 10). Fluorescence was measured at 470 nm emission and 395 nm excitation with a Varian Cary Eclipse Fluorescence Spectrophotometer (Agilent, Santa Clara, CA, USA). Data are expressed as nanomoles scopoletin per milligram of protein per min.

### 4.9. Measurement of Antioxidant Capacity

In the glomerular and proximal tubules, the antioxidant capacity was determined as previously reported [43]. For this purpose, 145 μL of 0.1 M phosphate buffer (pH 7.5 ± 0.5) was added to 50 μL of the sample. It was gently homogenized for 5 min on a MS3 digital vortex (IKA Works Inc., Wilmington, NC, USA) and 100 μL of the mixture was taken, to which 50 μL of 0.01 M copper II chloride was added. It was gently homogenized for 5 min (MS3 digital vortex, IKA Works, Inc., Wilmington, NC, USA). The mixture was allowed to stand for 10 min, and 50 μL of 0.01 M bathocuproine was added. It was gently homogenized for 5 min (MS3 digital vortex, IKA Works Inc., Wilmington, NC, USA) and analyzed spectrophotometrically at 490 nm excitation and 190 nm emission (Varian Cary Eclipse, Varian Inc., Mulgrave, Australia). Antioxidant capacity is expressed in μmol/mg protein.

### 4.10. Measurement of Malondialdehyde (MDA)

Capillary zone electrophoresis was used to determine malondialdehyde in glomerular and proximal tubule extracts [44]. For this, the samples were initially deproteinized with cold methanol (1:10) and subsequently with cold 10% (*v*/*v*) trichloroacetic acid (1:10). They were then centrifuged at 16,000× *g* at 10 °C for 15 min (Sorvall SR70, Thermo Scientific Inc., Urbana, IL, USA). The sample was filtered with 0.22 μm nitrocellulose filters (Millipore, Billerica, MA, USA). The filtrate was diluted with cold 0.1 M sodium hydroxide (1:10), and a cold Sep-Pak Classic C-18 cartridge (Waters, Urbana, IL, USA) pretreated with 10 mL of 100 mM ammonium citrate buffer (pH 2.5) was used to purify the sample for analysis. For this purpose, the P/ACE TM MDQ system (Beckman Coulter Inc., Fullerton, CA, USA) with UV–Vis detection by diode array was used. The analysis was performed at a voltage of −20 kV for 4 min, at 267 nm and 10 °C. The sample was injected under hydrodynamic pressure at 0.5 psi/10 s. The running buffer was 100 mM borates + 0.5 mM acetyl trimethyl ammonium bromide (pH 9.0). Malondialdehyde concentration is expressed in pmoles/mg protein.

### 4.11. Gelatin Zymography

Zymography was performed to identify MMP-2 and MMP-9 activity in renal tissue, and it was assayed using gelatin-substrate gels as previously described [45]. Briefly, 50 μg of non-denatured samples were mixed with non-denaturing sample buffer (2.5% SDS, 2% sucrose, and 0.01% phenol red) and separated in 8% native polyacrylamide gels co-polymerized with gelatin (1 mg/mL). Electrophoresis was carried out at 90 V for 2 h. The gels were rinsed twice in 2.5% Triton X-100 to remove SDS and then incubated in assay buffer (50 mM Tris–HCl, 150 mM NaCl, 10 mM CaCl2, and pH 7.4) at 37 °C for 48 h. Gels were fixed and stained with 0.25% Coomassie Brilliant Blue G-250 in 10% acetic acid and 30% methanol. Proteolytic activity was detected as clear bands on the blue background stain of undigested substrate in the gel at the expected location according to the molecular weights of MMP-9 and MMP-2. We use fetal bovine serum (FBS) as a positive control. Quantification was performed using ImageJ software V. 3.3.0 (National Institute of Health, Bethesda, MD, USA).

### 4.12. Statistical Analysis

Results were expressed as mean ± standard error of the mean (SEM). Differences between means from four groups were examined employing the one-way analysis of variance (ANOVA) followed by the Tukey post hoc test. Experimental data of capillary glucose concentrations before and during an OGTT were examined employing the two-way ANOVA followed by the Tukey post hoc test. Statistical significance was set at *p* < 0.05. Statistical analyses were performed with GraphPad Prism 9.0 (GraphPad Software, Inc., San Diego, CA, USA).

## 5. Conclusions

In conclusion, our results demonstrate that stimulation of PPAR-α reduced losses of claudin-2 in proximal tubules and claudin-5 in glomeruli by decreasing kidney oxidative stress and MMP-2 and MMP-9 activities. Improvements in claudin levels were associated with the restoration of renal function. These findings suggest that PPAR-α agonists are potential nephroprotective agents able to counteract glomerular and tubular dysfunction due to loss of TJ proteins in early stages of DN.

## Figures and Tables

**Figure 1 ijms-25-13152-f001:**
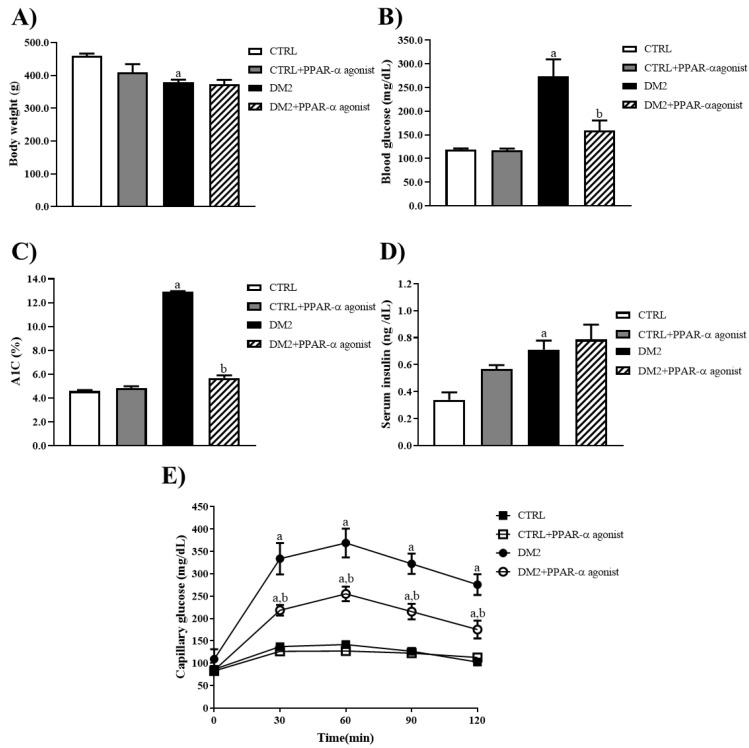
PPAR-α agonist treatment decreases hyperglycemia. (**A**–**C**) Body weight, blood glucose, and glycated hemoglobin (A1C) levels were monitored at 14 weeks after STZ administration. (**D**) Serum insulin levels at 14 weeks. (**E**) Capillary glucose concentration (mg/dL) plotted against time. The data represent mean ± SEM values (*n* = 4–6 rats evaluated per group). Data on (**A**–**D**) were analyzed by a one-way ANOVA, and data on (**E**) were analyzed by a two-way ANOVA followed by a post hoc Tukey test. a *p* < 0.05 versus CTRL, and b *p* < 0.05 versus DM2.

**Figure 2 ijms-25-13152-f002:**
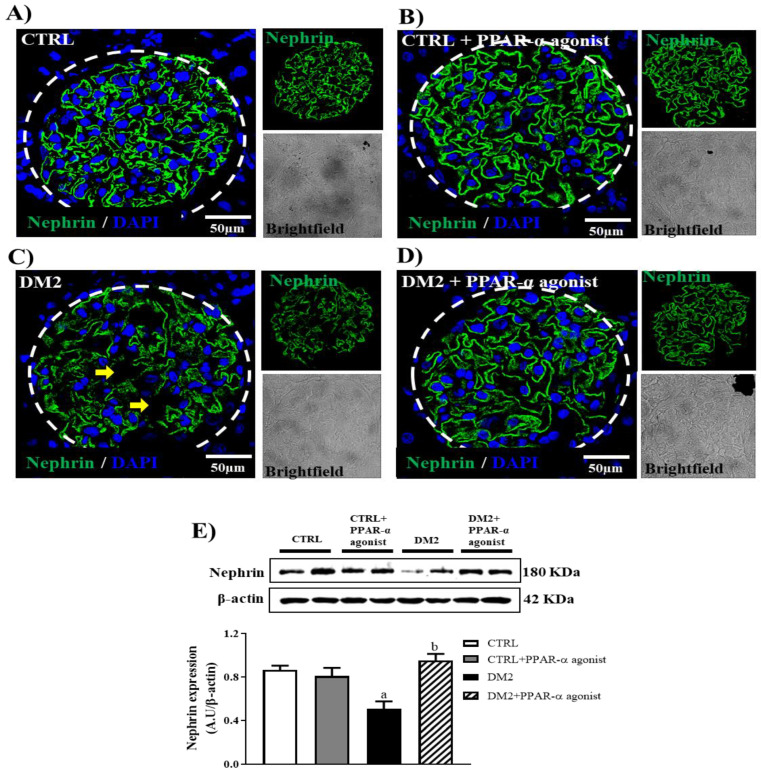
Nephrin expression is significantly upregulated in DN after treatment with PPAR-α agonists. Representative immunofluorescence images from kidney sections of four experimental groups showing nephrin expression in green and DAPI as a nuclear marker (**A**) CTRL, (**B**) CTRL + PPAR-α agonist, (**C**) DM2, and (**D**) DM2 + PPAR-α agonist. The yellow arrow shows that the continuous staining of nephrin disappeared in the DM2 group. (**E**) Western blot and relative quantification of nephrin against β-actin used as a loading control, from glomerular extracts of CTRL and DM2 groups in the presence or absence of PPAR-α agonist treatment. Bar = 50 μm. Dotted line delineates glomeruli. Data represent mean ± SEM values (*n* = 4–6 rats evaluated per group); a *p* < 0.05 versus CTRL, and b *p* < 0.05 versus DM2 from a one-way ANOVA followed by a post hoc Tukey test.

**Figure 3 ijms-25-13152-f003:**
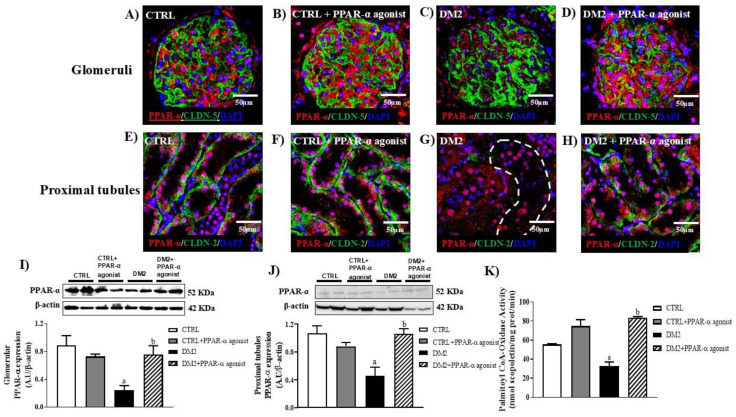
PPAR-α agonist treatment increases PPAR-α expression and activity in kidneys of type 2 diabetic rats. (**A**–**H**) Representative immunofluorescence images from kidney sections of four experimental groups showing PPAR-α expression in red, claudin-5 in glomeruli or claudin-2 in proximal tubules in green, and DAPI as a nuclear marker. (**I**,**J**) Western blot and relative quantification of PPAR-α against β-actin used as a loading control, from glomerular extracts and proximal tubules of CTRL and DM2 groups in the presence or absence of PPAR-α agonist treatment, respectively. (**K**) Activity of PPAR-α was measured as Palmitoyl CoA-Oxidase activity in the kidney cortex of every group. Bar = 50 μm. Dotted line in panel G delineates proximal tubule. Data represent mean ± SEM values (*n* = 4–6 rats evaluated per group); a *p* < 0.05 versus CTRL, and b *p* < 0.05 versus DM2 from a one-way ANOVA followed by a post hoc Tukey test. PPAR-α, peroxisome proliferator activated-receptor α; CLDN-2, claudin-2; CLDN-5, claudin-5.

**Figure 4 ijms-25-13152-f004:**
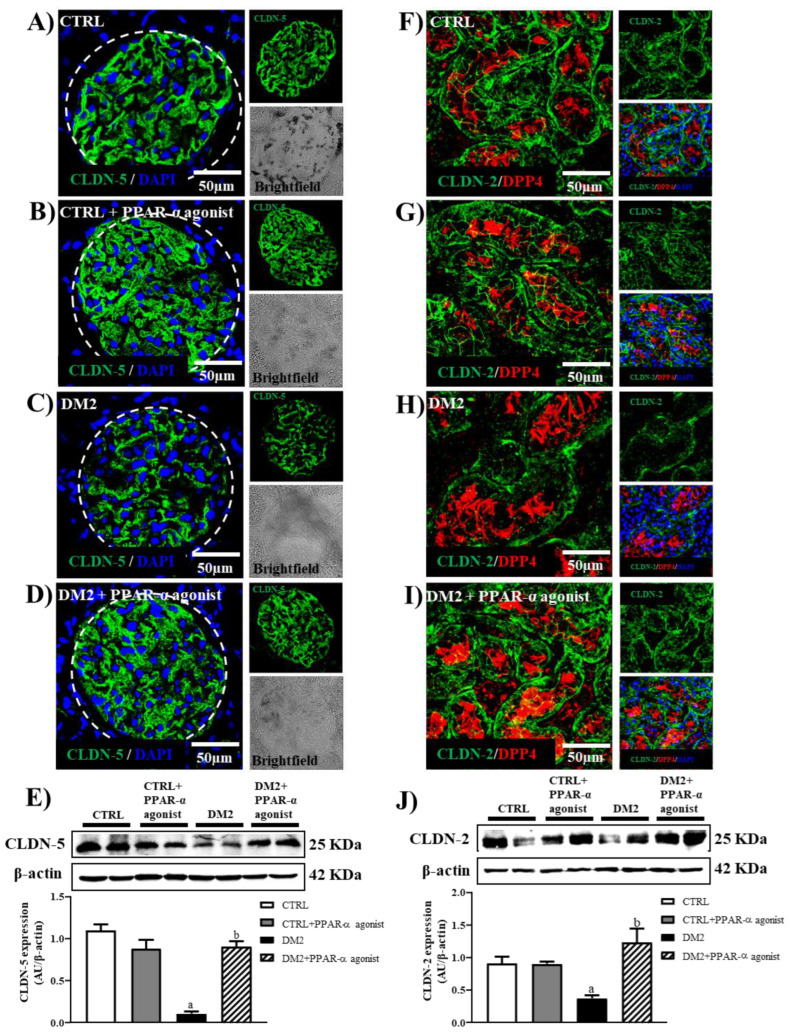
Effect of PPAR-α activation on renal claudin-5 and claudin-2 expression. Representative immunofluorescence images from kidney sections of four experimental groups showing claudin-5 expression in glomeruli in green and DAPI as a nuclear marker (**A**) CTRL, (**B**) CTRL + PPAR-α agonist, (**C**) DM2, and (**D**) DM2 + PPAR-α agonist. Representative double immunofluorescence images from kidney sections of four experimental groups showing claudin-2 expression in green, DPP4 as a marker of the apical membrane of the proximal tubule, and DAPI as a nuclear marker. (**F**) CTRL, (**G**) CTRL + PPAR-α agonist, (**H**) DM2, and (**I**) DM2 + PPAR-α agonist. (**E**) Western blots and relative quantification of claudin-5 against β-actin used as a loading control, from glomerular extracts of the four experimental groups. (**J**) Western blots and relative quantification of claudin-2 against β-actin used as a loading control, from proximal tubule extracts for the four experimental groups. Bar = 50 μm. Dotted line delineates glomeruli. Data represent the mean value ± SEM (*n* = 4–6 rats evaluated per group); a *p* < 0.05 versus CTRL, and b *p* < 0.05 versus DM2 from a one-way ANOVA followed by a post hoc Tukey test. CLDN-2, claudin-2; CLDN-5, claudin-5; DPP4, dipeptidyl peptidase 4.

**Figure 5 ijms-25-13152-f005:**
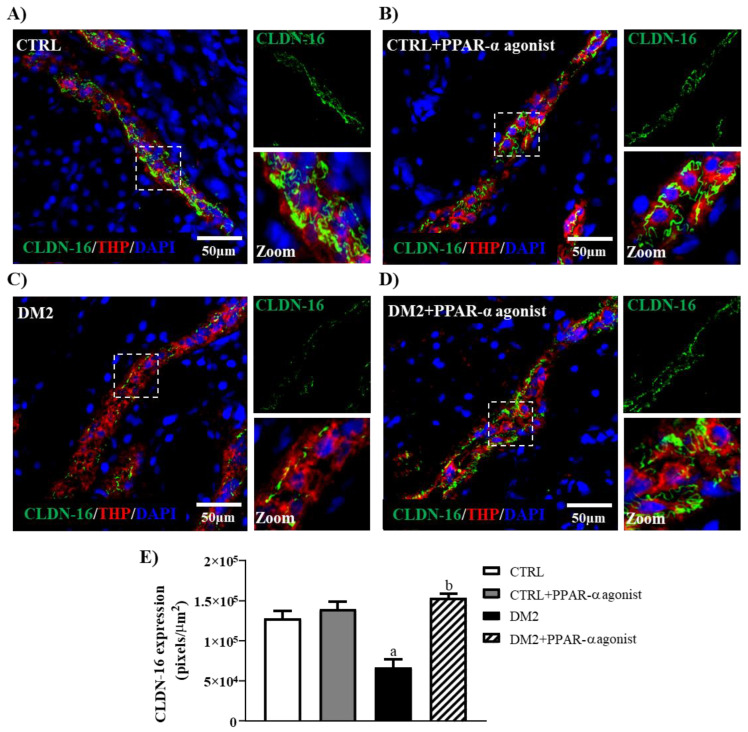
Effect of PPAR-α activation on renal claudin-16 expression in TAL. Representative double-immunofluorescence images showing claudin-16 in green, THP protein in red as a TAL marker, and DAPI as a nuclear marker in (**A**) CTRL, (**B**) CTRL + PPAR-α agonist, (**C**) DM2, and (**D**) DM2 + PPAR-α agonist kidney sections. (**E**) Quantification of the CLDN-16 expression. Bar = 50 μm. TAL boxed in white line was amplified for better appreciation. Data represent the mean value ± SEM (*n* = 4–6 rats evaluated per group); a *p* < 0.05 versus CTRL, and b *p* < 0.05 versus DM2 from a one-way ANOVA followed by a post hoc Tukey test. CLDN-16, claudin-16; THP, Tamm–Horsfall protein.

**Figure 6 ijms-25-13152-f006:**
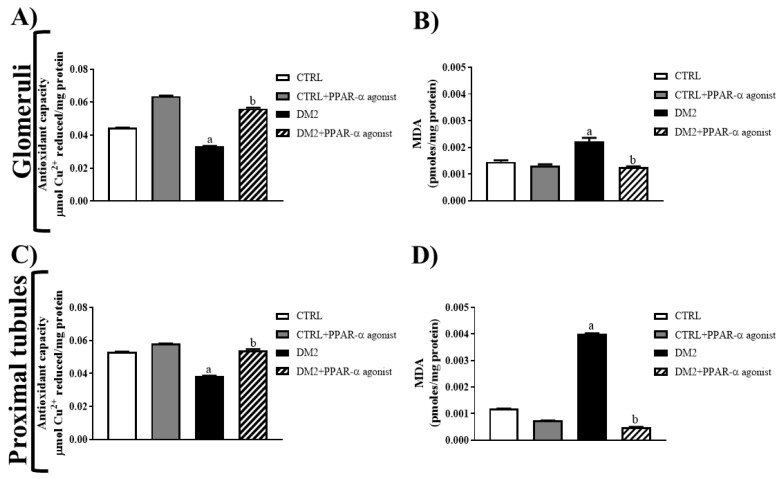
Antioxidant effect of PPAR-α agonist treatment in glomeruli and proximal tubules of diabetic rats. (**A**,**B**) Antioxidant capacity and MDA levels in isolated glomeruli, respectively. (**C**,**D**) Antioxidant capacity and MDA levels in proximal tubules, respectively. The bars represent the mean value ± SEM (*n* = 6 rats evaluated per group); a *p* < 0.05 versus CTRL, and b *p* < 0.05 versus DM2 from a one-way ANOVA followed by a post hoc Tukey test. MDA, malondialdehyde.

**Figure 7 ijms-25-13152-f007:**
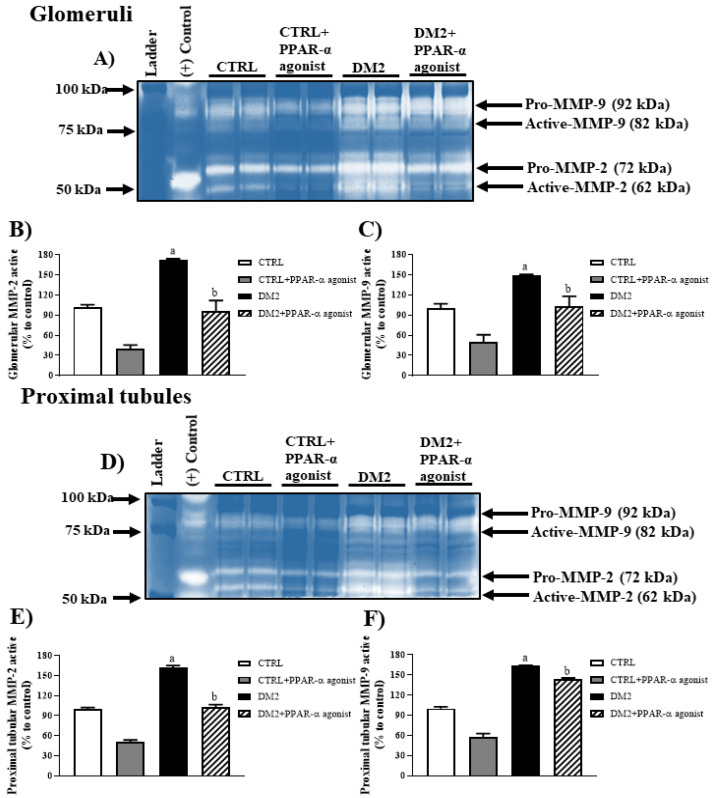
PPAR-α activation inhibits MMP-2 and MMP-9 activity in kidneys of diabetic rats. (**A**) Representative gelatin zymography of isolated glomeruli from every experimental group in the presence or absence of PPAR-α agonist. Both pro and active forms of MMP-2 and MMP-9 are denoted. (**B**,**C**) Densitometry analysis of MMP-2 and MMP-9 gelatinolytic activity in zymograms are shown as percentages of MMP-2 and MMP-9 active against the control in glomeruli extracts. (**D**) Representative gelatin zymography of proximal tubules from every experimental group in the presence or absence of PPAR-α agonist. (**E**,**F**) Densitometric analysis of MMP-2 and MMP-9 gelatinolytic activity in zymograms are shown as percentages of MMP-2 and MMP-9 active against the control in proximal tubule extracts. The bars represent the mean value ± SEM (*n* = 6 rats evaluated per group); a *p* < 0.05 versus CTRL, and b *p* < 0.05 versus DM2 from a one-way ANOVA followed by a post hoc Tukey test. (+) Control, positive control. MMP-2, matrix metalloproteinase 2; MMP-9, matrix metalloproteinase 9.

**Table 1 ijms-25-13152-t001:** Renal function parameters and Na^+^, K^+^, Mg^2+^, and Ca^2+^ handling by control and diabetic rats with and without PPAR-α agonist treatment.

	CTRL	CTRL + PPAR-α Agonist	DM2	DM2 + PPAR-α Agonist
Proteinuria (mg/24 h/100 g BW)	14.4 ± 0.5	11.0 ± 0.6	33.0 ± 2.5 **^a^**	22.0 ± 1.2 **^a,b^**
C_Cr_ (mL/min/100 gBW)	0.28 ± 0.02	0.28 ± 0.02	0.49 ± 0.04 ^**a**^	0.37 ± 0.03 **^b^**
FE_Na_ (%)	0.24 ± 0.03	0.36 ± 0.04	0.51 ± 0.07 ^**a**^	0.26 ± 0.06 **^b^**
FE_K_ (%)	12.49 ± 2.07	15.62 ± 2.77	27.23 ± 3.87 ^**a**^	7.91 ± 1.29 **^b^**
FE_Mg_ (%)	9.11 ± 1.36	8.49 ± 0.08	16.90 ± 3.32 ^**a**^	6.31 ± 0.37 ^**b**^
FE_Ca_ (%)	0.22 ± 0.04	0.24 ± 1.12	1.53 ± 0.45 ^**a**^	0.17 ± 0.03 ^**b**^

CCr, creatinine clearance; FENa, fractional excretion of sodium; FEK, fractional excretion of potassium; FEMg, fractional excretion of magnesium; FECa, fractional excretion of calcium. All data represent the mean ± SEM values (*n* = 4–6 rats evaluated per group). Statistical analysis was performed using a one-way ANOVA followed by a Tukey post hoc test; a *p* < 0.05 versus CTRL, and b *p* < 0.05 versus DM2.

## Data Availability

The original contributions presented in this study are included in the article/Appendix A. Further inquiries can be directed to the corresponding author.

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
