# Peer review of "Peroxisome Proliferator-Activated Receptor Alpha Stimulation Preserves Renal Tight Junction Components in a Rat Model of Early-Stage Diabetic Nephropathy"

_ijms, 2024, doi:10.3390/ijms252313152_

Round 1

Reviewer 1 Report

Comments and Suggestions for Authors

The minimum number of animals in each experimental group reduces statistical power. A limited sample size may reduce the reproducibility of the results, making them more susceptible to random variation. Furthermore, the study lacks power calculations, which would reveal whether the sample size was adequate to detect significant differences.
Clofibrate is evaluated as a single dose. A dose-response analysis, or multiple doses, may give more light on the therapeutic window and potential adverse effects.
Despite analyzing claudins 2, 5, and 16 as indicators of renal function, the study does not compare them to other common DN biomarkers such as albuminuria and eGFR, limiting their clinical use.

The study assesses tight junction proteins and oxidative stress indicators, but it does not consider overall renal outcomes or the potential links between these changes and renal function in a larger physiological context.

Reviewer 2 Report

Comments and Suggestions for Authors

Chronic hyperglycemia leads to kidney damage and diabetic nephropathy (DN), for which early treatment strategies are important. In this study, the authors showed that PPAR-α stimulation improved renal function in a rat model of type 2 diabetes by stabilizing tight junction proteins (claudin-2, claudin-5, and claudin-16), reducing oxidative stress, and decreasing MMP activity.

Major comments

1. There is no clear mechanistic pathway directly linking the stabilization of claudin-16 to PPAR-α activation.

2. This study relies on a single oxidative stress marker, but evaluating other oxidative stress markers could strengthen the findings.

Minor comments

Table 1; Does proteinuria mean albuminuria?

SEM in the graphs should be indicated by standard deviation (SD).

Reviewer 3 Report

Comments and Suggestions for Authors

This is a well thought out project that may be pertinent to the treatment of diabetic kidney disease. The manuscript can use editorial help and the authors need to address a few points in both the presentation of the data and in the discussion of the results.

Minor comments:

-Please use the proper abbreviations for proteins in the text, figure legends and figures.

Claudins are CLDN not Cl.

Dipeptidyl peptidase 4 is DPP4 not DppD.

-Please include a reference or a more detailed methodology for OGTT in the methods section.

Comments:

Section 2.1 and Fig.1.

-Please include information on renal function.

-The immunofluorescence microscopy images will be easier to evaluate if the staining was for PPAR-a only. Also, please make background lighter so the surrounding renal tissue can be visible (this recommendation applies to all IF images).

-The western blot shown in Section J is not convincing since there is a transfer problem with one of the DM2 lanes.

Section 2.2 and Fig.2.

-Results depicted in panels 2D and E suggest that PPAR-a agonist improves the response to insulin. This improved insulin sensitivity is well documented. In addition, PPAR-a is and inducer of Let-7c miRNA. This miRNA is involved in the regulation of insulin response, glucose homeostasis cell and oxidative injury, could the regulation of this miRNA by PPAR-a be the driver of renal protection observed in these studies? The authors should address this in the discussion.

Section 2.4 and Fig. 5.

-It would be a lot easier to decipher the zymography gels if the MMP9 and 2 results were separated. Also the graph legend should reas as "Percent of Control."

Section 2.5 and Fig. 6.

-DppD should be replaced with DPP4.

-The labeling of y-axis in panels E (b-actin not GAPDH) and J (b-actin not actina) need to be corrected.

Discussion.

-While these studies emphasize the role protective role of PPAR agonists in rats it is clear that the expression and activity of these receptors in humans vs rodents is very different. Specifically, nice with humanized PPAR receptor do not respond to the agonist in a similar fashion as wildtype mice. This point needs to be discussed.

-As I mentioned above, the authors should address the potential role of improved insulin response in the protection imparted by PPAR-a. This improved insulin sensitivity is well documented. In addition, PPAR-a is and inducer of Let-7c miRNA. This miRNA is involved in the regulation of insulin response, glucose homeostasis cell and oxidative injury, could the regulation of this miRNA by PPAR-a be the driver of renal protection observed in these studies? 

Comments on the Quality of English Language

The manuscript can use editorial help since there are quite a few sentences that are convoluted and hard to follow. 

Round 2

Reviewer 1 Report

Comments and Suggestions for Authors

Paper can be accepted in its present form.

Author Response

We would like to thank you for your time to follow this process and for your comments that improved significantly our manuscript.

Reviewer 2 Report

Comments and Suggestions for Authors

I do not recommend to use SEM. Otherwise, the authors replyed to my questions.

Author Response

(The authors gave the same response as above.)

Reviewer 3 Report

Comments and Suggestions for Authors

While almost all of my questions and concerns have been addressed, I am absolutely adamant that a better western blot picture for figure 3J needs to be provided. 

Author Response

Dear Reviewer, 

Thank you very much for taking the time to re-review this manuscript. Please find the detailed responses below and the corresponding revisions/corrections highlighted/in track changes in the re-submitted files.

Round 3

Reviewer 3 Report

Comments and Suggestions for Authors

All my concerns have been addressed.